# Consistency Guided Diffusion Model with Neural Syntax for Perceptual Image Compression

## ABSTRACT

Diffusion models show impressive performances in image generation with excellent perceptual quality. However, its tendency to introduce additional distortion prevents its direct application in image compression. To address the issue, this paper introduces a Consistency Guided Diffusion Model (CGDM) tailored for perceptual image compression, which integrates an end-to-end image compression model with a diffusion-based post-processing network, aiming to learn richer detail representations with less fidelity loss. In detail, the compression and post-processing networks are cascaded and a branch of consistency guided features is added to constrain the deviation in the diffusion process for better reconstruction quality. Furthermore, a Syntax driven Feature Fusion (SFF) module is constructed to take an extra ultra-low bitstream from the encoding end as input, guiding the adaptive fusion of information from the two branches. In addition, we design a globally uniform boundary control strategy with overlapped patches and adopt a continuous online optimization mode to improve both coding efficiency and global consistency. Extensive experiments validate the superiority of our method to existing perceptual compression techniques and the effectiveness of each component in our method.

## CCS CONCEPTS

• Computing methodologies → Image compression.

## KEYWORDS

Image compression, generative model, denoising diffusion model, neural syntax

## 1 INTRODUCTION

Image compression technology plays a pivotal role in diverse fields, *e.g.*, multimedia, communications, and computer vision. Its objective is to efficiently minimize the storage space and bandwidth requirements of digital images for their efficient storage and transmission, while preserving the main content of the original images and maintaining the visual quality. In today's digital age, with the ever-increasing demand for high-resolution and high-quality images from multimedia devices, the challenge of managing storage and bandwidth resources has become paramount, and the demand for more efficient and high-performance image compression methods is also growing rapidly.

*ACM MM, 2024, Melbourne, Australia*

© 2024 Copyright held by the owner/author(s). Publication rights licensed to ACM.
ACM ISBN 978-x-xxxx-xxxx-x/YY/MM
https://doi.org/10.1145/nnnnnnn.nnnnnnn

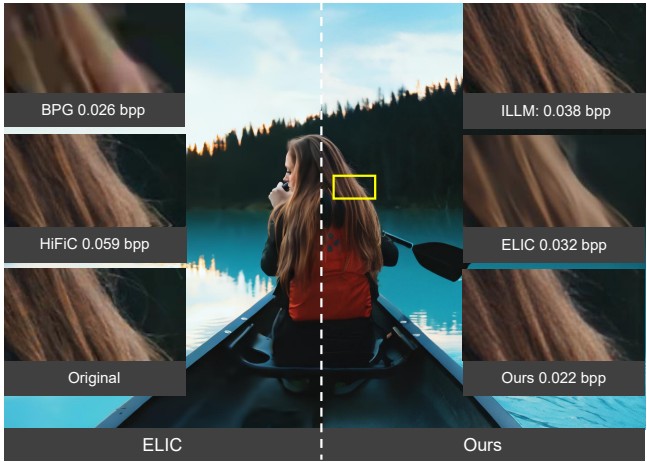

**Figure 1: Visual comparisons of different methods. The patch is cropped from *roberto-nickson-48063.png* from CLIC professional dataset [50]. Compared to previous methods, our method achieves better or competitive perceptual quality at a lower bitrate. In particular, our method achieves similar perceptual quality using about half bitrate compared to the milestone HiFiC [38]. [*Zoom in for best view*]**

Over the past decades, conventional image compression techniques like JPEG [52], BPG [5] and JPEG2000 [37] have become the common choice in image processing. These methods or standards exhibit excellent encoding capabilities, adopting the route of transform/hybrid coding framework for Rate-Distortion Optimization (RDO) with key modules such as transformation, quantization, and entropy coding, while being complemented by additional complex predictive modes. Moreover, numerous efforts have been made to improve the RDO of each input image by projecting the image signals into a manually designed specific subspace for more compact representations, *e.g.* intra-prediction based on various directions [52]. Nonetheless, these optimizations rely on human design, lacking the capability for global optimization. As the number of manually designed strategies continues to grow, the framework of the compression model becomes increasingly complicated, gradually revealing its performance bottlenecks.

In recent years, deep learning technologies have made remarkable advancements, prompting many researchers to explore image compression methods with the immense capabilities of neural networks [4, 18, 24, 35, 53]. By leveraging vast datasets to train neural networks, these methods excel in discovering the latent relationships and underlying structures embedded within image data. As a result, they attain a remarkable compression efficiency, while ensuring minimal distortion, thereby surpassing traditional techniques in terms of performance. While these methods have achieved a

notable level of image compression and reconstruction, they gradually encounter performance bottlenecks: further improving the compression ratio resulting in significantly degraded quality.

Some researches [6, 55] indicate that the distortion of images does not align with human perception of subjective image quality. Due to the inherent trade-off between image quality and storage efficiency, perceptual image compression techniques [38, 42, 56] are proposed. These technologies strive to protect the quality in terms of human visual perception rather than focusing on fidelity measurement. To enhance the subjective visual quality of images, many generative models [17, 32, 51] skill at producing visually appealing details such as GANs are seamlessly integrated into image compression methods [38, 42, 58] resulting in a significant improvement in perception quality. However, the development of these methods is constrained by the inherent limitations of generative models, including the frequent lack of diversity in the images produced by GAN models and so on, thereby posing significant challenges in their further advancement.

In recent years, diffusion models [21, 34, 48] have emerged as a powerful tool in the field of image generation, exhibiting remarkable capabilities in producing images with exceptional perceptual quality. These models, developed according to the formulation of diffusion processes, have demonstrated their ability to capture intrinsic details and generate realistic images. However, despite their remarkable success in image generation, it has been established in many practices that vanilla diffusion models tend to reconstruct images with richer visual details at a cost of significantly impaired fidelity [45], due to the random nature of the process of progressive adding or removing noise. Applying the diffusion models directly to the image compression task may take on risks of a synchronous drop of both visual quality and fidelity. Thus, until now, how to apply diffusion models to image compression remains under-explored.

To address the issue of utilizing the power of diffusion models while avoiding their generated artifacts, this paper proposes to regularize the diffusion models with global consistency guidance. In detail, we propose a novel approach called the **Consistency Guided Diffusion Model (CGDM)**, which incorporates additional consistent guidance into the network structure of the diffusion model. This approach aims to constrain deviations in the diffusion process for improving the quality of the reconstructed image. Furthermore, we propose a Syntax driven Feature Fusion (SFF) strategy. This strategy encodes an additional ultra-low bitstream obtained from the encoding stage, providing semantic prior information about the image. By leveraging this prior information, we can reduce the ambiguity in the inference target during the post-processing phase, leading to more accurate and faithful reconstructions. To achieve the same objective of reducing randomness in the diffusion process and effectively leveraging image semantic information, we apply a globally consistent edge control strategy into our model's inference phase. Additionally, we adopt a continuous online optimization mode to further enhance the model's performance. These efforts not only reduce the stochasticity associated with diffusion processing but also entourage the model to capture rich semantics from the input images, thereby leading to improved overall performance.

Our contributions are summarized as follows:

- We develop a Consistency Guided Diffusion Model (CGDM) for perceptual image compression, which incorporates an additional consistent guidance with a diffusion-based network, aiming to constrain the deviation in the diffusion process for learning richer detail representations with less fidelity loss.
- We devise a Syntax driven Feature Fusion (SFF) module that takes an extra ultra-low bitstream from the encoding end as input, guiding the adaptive fusion of information from the two branch.
- We design a globally uniform boundary control strategy, and adopt continuous online optimization mode to further improve both coding efficiency and global consistency.

Experimental results show that our proposed method achieves a BD-rate [16] savings of 9.227% in perception and 6.251% in distortion compared to the current state-of-the-art perceptual image compression method ILLM [42].

## 2 RELATED WORKS

### 2.1 Generative Model

Generative models aim to learn the overall distribution of data and generate data within the same distribution, which have been a central focus of research in recent years, leading to significant advancements in multimedia generation and processing. With the rapid development of deep learning, one of the milestone and the most notable generative models is the Generative Adversarial Networks (GANs) [17], which consist of two competing networks, a generator and a discriminator. The adversarial training process results in highly realistic and diverse generations. Subsequent work built upon GANs such as Conditional GAN [41], StyleGAN [25, 26] further enhances its ability to generate high-quality images based on given conditions. Beyond GANs, there has also been a surge of interest in other types of generative models, including autoregressive models [15], variational autoencoders (VAEs) [29], normalizing flow models [11], energy-based models [12], score-based model [49], flow-based model [28] and so on.

Recently, diffusion models [21, 34, 48] have emerged as powerful generative models that define the forward and reverse diffusion processes for data noise addition and removal, respectively. Their generations often exhibit superior quality and diversity, and there have been many studies attempting to use diffusion-based models for image generation [9, 59], enhancement [13, 36, 43, 57] and so on. In our work, we apply the diffusion-based generation model to image compression, and obtain the image with higher quality through the guidance of a semantic stream.

### 2.2 Learned Image Compression

With the significant advancements in deep learning, recent years have witnessed deep learning based image compression methods outperforming classical methods in striking a balance between bit rate and reconstruction quality. Initially, Ballé *et al.* [2, 3] pioneered the utilization of neural network to establish lossy image compression autoencoders, sparking a surge in learning-based image compression methods [39, 40]. In addition to transformations, numerous studies have focused on entropy coding of latent representations based on learned probability models, including hyperpriors [4] and context models [7, 33]. Furthermore, the employment of Gaussian

Mixture Models and attention-based modules in transformations has further enhanced image compression performance [8].

Facing the trade-off between image quality and storage efficiency, a range of perceptual image compression methods have been proposed, which aim to enhance the perceptual quality of compressed images and align them more closely with human perception. Agustsson *et al.* [1] introduced the concept of using GANs [17] as decoders for image compression. This approach allows for the generation of reconstructed images with rich details. Subsequently, He *et al.* [19] further enhanced these GAN-based methods by incorporating advanced perception models. Recently, with the great success of diffusion models, some efforts [22, 56] have been made to study perceptual image compression. However, as we have stated previously, due to the lack of fidelity caused by the uncertainty of the diffusion process, this area needs to be further explored. In this paper, we propose a solution for this issue through an additional consistent guidance and a neural syntax driven strategy.

## 3 METHOD

In this part, we first describe general information of diffusion models while outlining our motivations in Section 3.1, followed by a detailed elaboration on our proposed consistency guided diffusion model in Section 3.2. Then, we further propose our syntax driven feature fusion module in Section 3.3. The globally uniform boundary control strategy and continuous online optimization during inference is introduced in Section 3.4. Finally, our training strategy is described in Section 3.5.

### 3.1 Preliminaries and Motivations

We start with the characteristic analysis of the diffusion model. As a generative model, diffusion models have been demonstrated to effectively create images with excellent perceptual quality by leveraging a conditional model that incorporates latent features.

Simultaneously, there are numerous works [13, 36, 43, 57] that employ the diffusion models as a post-processing or enhancement module. Generally, these works utilize the degraded image $\tilde{x}$ as a condition and construct a conditional model that aims to learn the data distribution $p(x|\tilde{x})$ through a fixed multi-step chain of length $T$. The diffusion process is defined by a forward process $q$ through adding Gaussian noise. Formally, the distribution of the forward process can be expressed as:

$$q(x_t|x_0) = \mathcal{N}(x_t; \alpha_t x_0, \sigma_t^2 \mathbf{I}),$$
$$q(x_T) = \mathcal{N}(x_T; \mathbf{0}, \mathbf{I}), \tag{1}$$

where $\alpha_t$ and $\sigma_t^2$ are hyper-parameter functions of $t$ [34].

Meanwhile, the inference process can be conducted as a reverse process from Gaussian noise $q(x_T) \sim \mathcal{N}(\mathbf{0}, \mathbf{I})$ to a target $x_0$, which can be expressed as:

$$p(x_T) = \mathcal{N}(x_T|\mathbf{0}, \mathbf{I}),$$
$$p(x_{t-1}|x_t, \tilde{x}) = \mathcal{N}(x_{t-1}|\mu_\theta(\tilde{x}, x_t, t), \sigma_t^2 \mathbf{I}), \tag{2}$$

where the $\mu_\theta(\tilde{x}, x_t, t)$ denotes the mean value of the conditional distribution $p_\theta(x_{t-1}|x_t, \tilde{x})$, and the diffusion model is trained to learn the conditional distributions by parametric approximation to the distribution. For the neural network, mostly existing diffusion post-processing frameworks directly feed the condition $\tilde{x}$ and noise

$x_t$, along with the timestamp $t$, into the U-Net backbone, similar to the vanilla DDPM and output the predicted noise $\hat{\epsilon}_t$ at each step.

However, there are two notable issues with this paradigm:

- This approach often leads to the final reconstructed image $x_0$ deviating from the condition $\tilde{x}$, amplifying the distortion of $\tilde{x}$ and results in simultaneous degradation of both fidelity and perceived quality.
- As the condition $\tilde{x}$ represents a degraded image with some information loss, there can exist multiple images that lead to the same $\tilde{x}$. That means, the optimized probability distribution target $p(x|\tilde{x})$ is ambiguous, making it challenge to ensure that the reconstructed image is more similar to the original image.

In our work, we aim to adopt a novel framework to address both of these issues:

- For the first issue, we propose to incorporate an additional consistent guidance into the network structure of the diffusion model, called consistency guided diffusion model, constraining the deviation in the diffusion process and improving the quality of the reconstructed image.
- For the second issue, we propose a syntax driven feature fusion strategy to encode an additional ultra-low bitstream $s$ from the encoding stage to provide semantic prior information of the image, thereby alleviating the ambiguity in the inference target of post-processing.

In the following sections, we describe our method in detail.

### 3.2 Consistency Guided Diffusion Model

On a high level, our compression process consists of two parts, an end-to-end image compression model and a diffusion-based post-processing model called consistency guided diffusion model, with structure shown in Fig. 2.

**End-to-end image compression model.** Firstly, we utilize a standard end-to-end image compression network to perform lossy compression on the original image $x$, obtaining an image with some detailed information lost $\tilde{x}$:

$$\tilde{x} = D(Q(E(x))), \tag{3}$$

where $Q()$ means quantizer and $E(), D()$ represents the pre-trained autoencoder. Here, we utilize the recently proposed ILLM architecture [42] as the end-to-end autoencoder, which is the current state-of-the-art perception-oriented end-to-end compression method.

**Diffusion-based post-processing model.** Our diffusion-based post-processing model follows the encoder-decoder architecture with skip connections [44] as the denoising model similar to [14], which includes two encoders and one decoder.

The upper branch in Fig. 2 encodes the noisy image into $N$ multi-resolution diffusion feature maps $d_i$ with different scales, where $N$ is the depth of the U-Net backbone and $i \in \{1, ..., N\}$, while the lower branch extracts feature maps $e_i$ of corresponding scales from the image $\tilde{x}$. Then, we introduce a syntax driven feature fusion module guided by an ultra-low semantic information bitstream $s$ in the decoder part of the U-Net, which is expounded in the following section. Initially, the feature $d_N$ and $e_N$ are fused to obtain the feature $u_N$. Then, at each layer, the corresponding layer's feature $u_{i+1}$ and $e_i$ are adaptively fused to produce $u_i$, which can

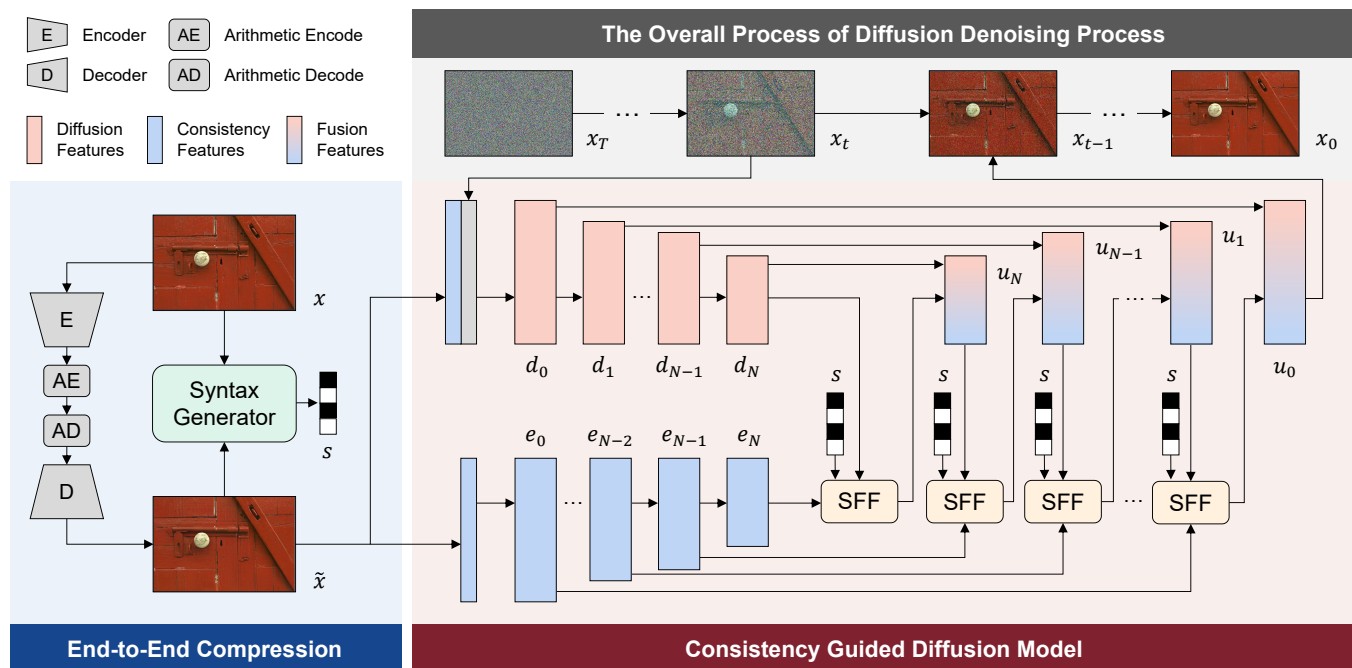

**Figure 2: The entire framework of our proposed method. For an image $x$ to be encoded, we first perform lossy compression using a standard end-to-end image compression network, resulting in an output degraded image $\tilde{x}$. Then, we extract a syntax vector from the original image $x$ using a syntax generator. This syntax vector is then used to guide the fusion of consistency features $e$ and diffusion features $d$ in a *Consistency Guided Diffusion Model with Neural Syntax*. After a complete diffusion process, we obtain a higher-quality reconstructed image $x_0$. The consistent guidance architecture, neural syntax driven mechanism lead the diffusion model to stably reconstruct high-quality images, making the final output excellent in terms of perception and fidelity.**

be expressed as:

$$u_i = \text{SFF}_i(u_{i+1}, e_i, s), \tag{4}$$

where $\text{SFF}_i$ means the syntax driven feature fusion module and $s$ is a compact syntax vector. This process is repeated layer by layer, with adaptive fusion and upsampling of features performed at each step, ultimately generating the predicted output noise.

By taking this approach, during the denoising diffusion process of the diffusion post-processing model, we consistently inject a constant guidance feature derived from the degraded image, guiding its inference process to stay close to the conditional distribution $\tilde{x}$. This approach enables the final output to enhance the perceptual quality while maintaining the similarity to the original image, achieving a better trade-off between fidelity and perceptual quality.

### 3.3 Syntax Driven Feature Fusion

As we mentioned in section 3.1, the target optimized probability distribution of the post-processing module $p(x|\tilde{x})$ is ambiguous. To mitigate the ambiguity to ensure the reconstructed image $x_0$ similar to original image $x$, we proposed to send a compact syntax vector extracted from the original image $x$ and decoded image $\tilde{x}$ which costs ultra-low bitstream to provide the syntax information of the original image. Based on this idea, inspired by [53], we encode the syntax information of the original image into a compact and discrete one-dimensional vector by a syntax generator which serve

as a dynamic convolution kernel in the syntax driven feature fusion module. By decoding the syntax vector into dynamic convolution kernels and performing convolution operations on the features to be fused, syntax information is transmitted in a neural representation-like manner. The structure of the syntax generator module and syntax driven feature fusion module is illustrated in Fig. 3.

**Syntax Generator.** The syntax generator's structure follows the design of [53], which contains a multi-scale network on the basis of hyper-priors entropy model [4, 23]. During syntax extraction, the features at each scale are globally average pooled and concatenate to a compact one-dimensional vector. This approach effectively utilizes multi-scale information while ensuring global consistency of the semantic information.

**Syntax Driven Feature Fusion.** The syntax driven feature fusion module takes the features $d_i, e_i$, the obtained syntax vector $s$, and the timestamp $t$ of the current step as inputs. After getting the syntax vector $s$ and timestamp $t$, we concatenate them and utilize a fully connected network to map them to two convolutional kernels $W_e^i, W_d^i$. These two sample-adaptive dynamic convolutional kernels separately perform convolution on the two input features, achieving adaptive fusion of features at each layer:

$$u_i = W_e^i * e_i + W_d^i * d_i, \tag{5}$$

where $*$ denotes convolution. Since the semantic vector used for generating these two convolutions is highly dependent on the input original image, the fusion process can effectively capture the

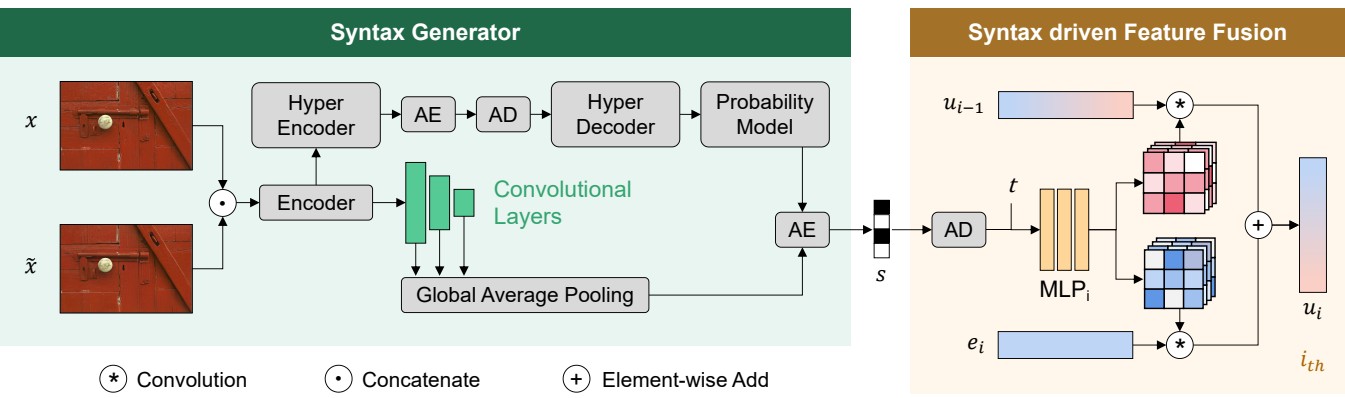

**Figure 3: The structure of Syntax Generator and Syntax driven Feature Fusion module (SFF). The Syntax Generator module is responsible for extracting global syntax vectors from images during the encoding process, while the syntax driven feature fusion module adaptively integrates consistency features with diffusion features within the diffusion framework.**

characteristics of the original image, enabling the more adaptive fusion of the two streams of features during the generation process to obtain a better reconstructed image.

### 3.4 Inference Time Optimization

Furthermore, we propose that due to the excellent sample adaptability of our method, a more refined design during the inference process can more fully tap into the performance potential of our proposed method and achieve better performance. Specifically, during the inference process, we employ a globally uniform boundary control strategy and a sample-adaptive continuous online optimization mechanism for different resolutions and styles of images to be compressed, respectively, to further enhance the performance.

**Globally Uniform Boundary Control.** During model training, we used patches of a fixed size. However, directly feeding images of different resolutions into the diffusion model during inference can cause performance degradation due to distribution discrepancies. Therefore, we chose to use a tiling approach to adapt to images of arbitrary resolutions. To mitigate the potential block artifacts that may arise from piece-by-piece tiling, we employ the following two strategies. Firstly, we overlap the patches with surrounding ones to a certain extent, so that the pixels at the edge position are predicted by multiple patches, which makes the transition of the edge position smoother.

In addition, we observe that the initial noise $x_T$ of the diffusion model can be viewed as the boundary condition of the diffusion ordinary differential equation, which affects the stylistic characteristics of the sampled images [36]. Recognizing that a complete image should exhibit consistent stylistic features, we set the initial noise for all patches to a fixed distribution. This ensures consistent image style, further mitigates block artifacts, and enhances the performance of encoding and decoding images at arbitrary resolutions. Specifically, for a given patch size, we commence by randomly selecting a boundary condition (just a Gaussian noise). Subsequently, we maintain this noise as a uniform boundary condition and tile it across the entire image, resulting in a boundary condition $x_T$, that spans the entire image. Using $x_T$ as the starting, we then employ

our CGDM to initiate the diffusion process. Through this process, we ultimately reconstruct a high-quality image, X0, that exhibits a stable and consistent style.

**Sample-Adaptive Continuous Online Optimization.** Similar to [35, 53], the approach of using a syntax generator to extract global syntax information from images inherently brings the potential for online optimization in the encoding phase during inference. This process is analogous to the mode decision process in traditional hybrid coding frameworks, where the best mode is selected from a discrete set of candidates. However, employing iterative optimization allows for continuous selection of the best option from an infinite set, greatly enhancing the flexibility of this online optimization strategy.

Specifically, during the inference process for each image, we iteratively optimize the encoder parameters of the syntax generator on randomly selected patches. This enables the generator to produce syntax vectors that more closely align with the image's semantics. For the iterative optimization process during inference, we set an optimization objective consistent with the fine-training process, which is detailed in the next section.

### 3.5 Training Strategy

Following the training strategy proposed by [54], we use a coarse-to-fine two-stage training strategy. Training at the coarse level aims to train the diffusion model to constrain noise, while training at the fine level further optimizes the diffusion model to further enhance the model performance by constraining the sampled clean images with fixed steps and corresponding ground truth ones.

Our coarse training is analogous to existing conditional diffusion models, which aim to estimate noise. The difference is, that we introduce an additional bitrate control term to constrain the bit cost of syntax vectors. Consequently, the loss function for the coarse training is as follows:

$$\mathcal{L}_{coarse} = \mathbb{E}_{t\sim[1,T],x_0,\epsilon_t} \|\epsilon_t - \mu_\theta(x_t,\tilde{x},s,t)\|_2 + \lambda_c R, \quad (6)$$

where $\mu_\theta$ denotes our noise prediction network CGDM, $\epsilon_t$ means the adding noise at step t in the forward process, $R$ means the

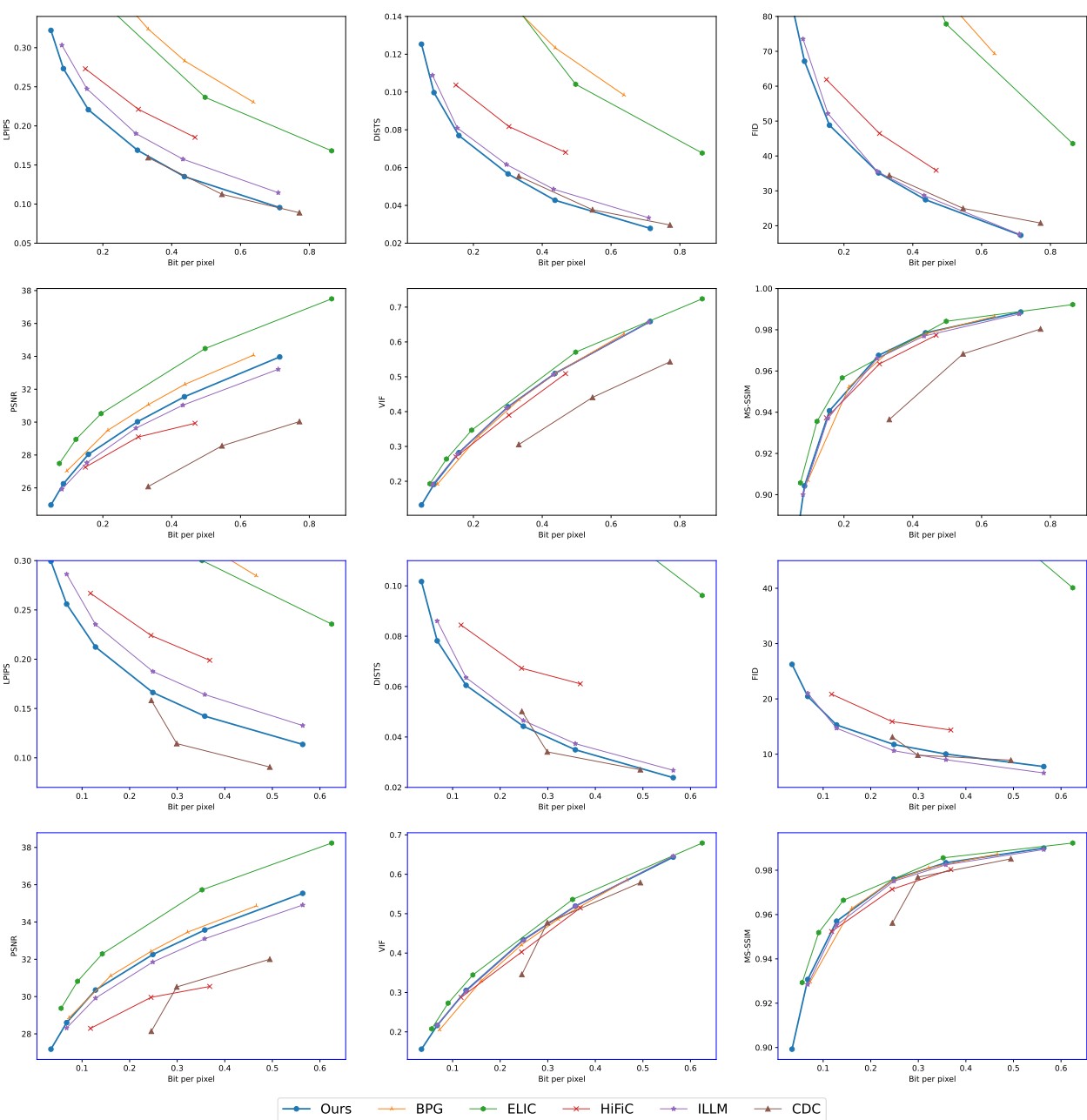

**Figure 4: Tradeoffs between bitrate (x-axes, in bpp) and different metrics (y-axes) for various models tested on Kodak and CLIC. We consider both perceptual (LPIPS, DISTS, FID) and distortion metrics (PSNR, VIF, MS-SSIM). The upper 2 rows(black frame) are the performance on Kodak datasets and the lower 2 rows (blue frame) are on CLIC professional dataset.**

bitrate and $\lambda_c$ is the hyper-parameter to trade-off between rate and distortion.

During the fine training stage, we fixed the sampling strategy to the 9-step DDIM sampling and imposed constraints on the generated sampled images $\hat{x}_0$ to compensate for the unsatisfactory results from the coarse training. The loss function we used for the

constraints is as follows:

$$\mathcal{L}_{fine} = \lambda_d \mathcal{L}_d(\hat{x}_0, x) + \lambda_p \mathcal{L}_p(\hat{x}_0, x) + \lambda_f R, \quad (7)$$

where $\mathcal{L}_d$ denotes the distortion loss and we use L1 loss during training, $\mathcal{L}_p$ denotes the perception loss and we utilize three kinds of perceptual objective functions DISTS [10], Alex-based [31] and VGG-based [47] LPIPS [60] in the experiments. $\lambda_d$, $\lambda_p$ and $\lambda_f$ are the hyper-parameters.

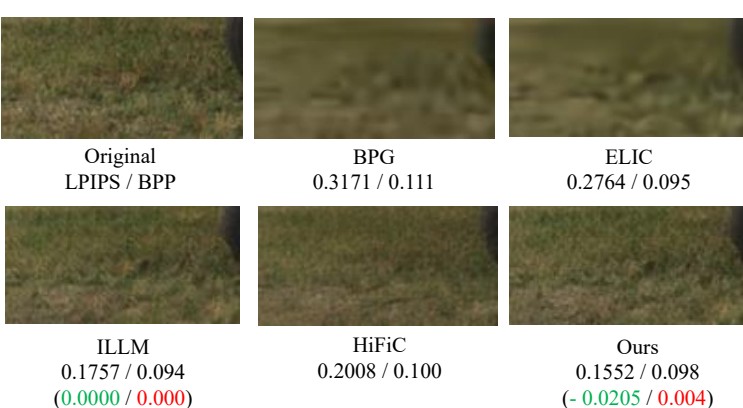

Original
LPIPS / BPP

BPG
0.3171 / 0.111

ELIC
0.2764 / 0.095

kodim20.png

ILLM
0.1757 / 0.094
(0.0000 / 0.000)

HiFiC
0.2008 / 0.100

Ours
0.1552 / 0.098
(- 0.0205 / 0.004)

Figure 5: Visual comparisons with state-of-the-art methods on Kodak dataset. We provide further analysis that focuses on subjective results in the main text. As can be seen, compared to the baseline used in our method (ILLM), we achieve a significant improvement in subjective performance at the cost of extremely low additional bitstreams. [*Zoom in for best view*]

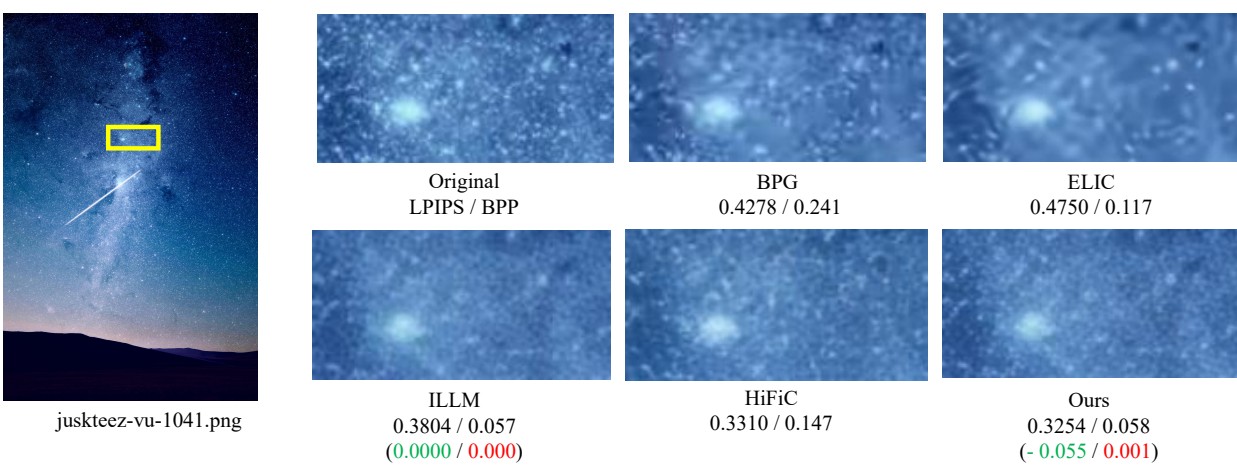

Original
LPIPS / BPP

BPG
0.4278 / 0.241

ELIC
0.4750 / 0.117

juskteez-vu-1041.png

ILLM
0.3804 / 0.057
(0.0000 / 0.000)

HiFiC
0.3310 / 0.147

Ours
0.3254 / 0.058
(- 0.055 / 0.001)

Figure 6: Visual comparisons with state-of-the-art methods on CLIC dataset. [*Zoom in for best view*]

## 4 EXPERIMENTS

### 4.1 Implementation

**Network Implementation.** We implement our diffusion model based on the architecture of [14] with fewer parameters. In addition, in order to further reduce the video memory consumption, we also removed the self-attention module. The detailed structure and hyperparameters of the networks are shown in the supplementary material.

**Training Details.** We utilize the DIV2K [1] dataset as our training dataset, which comprises 800 natural images with an average resolution of 2K. To enable our model to adapt to images of various resolutions, we perform downsampling on the images to half their original resolution, serving as an augmentation of the training data. During the training process, we randomly crop $256 \times 256$ patches from each image.

Our training process uses the Adam optimizer [27] and the learning rate is set to $1 \times 10^{-4}$. We trained 6 models with different

compression rates using different bit rates end-to-end compression model parameters. The hyper-parameter $\lambda_c$ on the coarse training stage is set to 100 and the hyper-parameter $\lambda_d$, $\lambda_p$ and $\lambda_f$ are set to 1, 0.3 and 20 separately on the fine training stage. Each model is trained for 38k iterations on the coarse training stage and 32k iterations on the fine training stage.

**Inference Details.** During inference, the patch size used is $256 \times 256$, and the overlap range is 8 pixels close to the edge. Our method applies the continuous mode decision on inference. For each image, based on the pre-trained network weights, we additionally employ the Adam optimizer with a learning rate $5 \times 10^{-5}$ to finetune the encoder for 250 iterations, and the optimization target is the same as $\mathcal{L}_{fine}$.

**Evaluation Protocol.** We evaluated our method on the Kodak image dataset [30] and the professional subset of the CLIC validation dataset [50]. The Kodak image dataset consists of 24 images,

**Table 1: Average BD-rate for different methods on both CLIC and Kodak datasets anchored on ours method.**

| Datasets | Kodak | | | CLIC | | |
|---|---|---|---|---|---|---|
| Methods | Distortion | Perception | Average | Distortion | Perception | Average |
| HiFiC [38] | 14.6366 | 45.1822 | 29.9094 | 43.4935 | 113.7039 | 78.5987 |
| ILLM [42] | 5.5011 | 11.4812 | 8.4912 | 6.9765 | 14.1819 | 10.5792 |
| CDC [56] | 52.0296 | 0.5460 | 26.2878 | 65.2966 | -21.5621 | 21.8673 |
| ELIC [18] | -31.6699 | 77.8194 | 23.0748 | -22.2319 | 2288.3859 | 1133.0770 |
| BPG [5] | -1.5526 | 84.3209 | 41.3842 | 3.0902 | 4872.2267 | 2437.6585 |
| Ours | — | — | — | — | — | — |

each with a resolution of $768 \times 512$. The CLIC professional validation dataset comprises 41 images with higher resolutions of about $1800 \times 1200$. Evaluating on it demonstrates the performance of our method on images with higher resolutions.

To demonstrate the superiority of our method in terms of distortion and perceptual quality, we utilized a set of diverse metrics. For distortion, we employ PSNR, VIF [46], and MS-SSIM [55]. And for perceptual quality, we used VGG-based [47] LPIPS [60], DISTS [10], and FID [20]. The R-D curves and BD-rate [16] on different evaluation metrics are illustrated to compare different methods and settings.

## 4.2 Quantitative Comparison

We compared our method with existing conventional transform-based methods BPG [5], end-to-end learning-based image compression methods optimized for MSE like ELIC [18], and image compression methods optimized for perceptual quality including HiFiC [38], ILLM [42], and CDC [56]. Fig. 4 presents the R-D curves of various metrics on the CLIC and Kodak datasets for our proposed method and comparison methods. Evidently, our approach demonstrates superior performance across different perceptual measures compared to other perceptual image compression methods, meanwhile achieving a more favorable distortion effect. Among the comparison methods, only CDC surpasses our approach in terms of the perception metrics. However, its performance in the distortion metrics was significantly inferior to other methods, resulting in an overall performance that remained below ours. This underscores the excellent balance our method achieves between fidelity and perceptual quality, highlighting its superiority.

To provide a more intuitive comparison of the overall performance of our method with other benchmark methods across all evaluation metrics, we computed the BD-rate [16] for each indicator. Using our method as the anchor, Table 1 presents the average BD-rate achieved by each method across all distortion and perception metrics on both CLIC and Kodak datasets anchored on our method. It is evident from the table that the overall performance of all other methods under all evaluation indicators falls below that of our proposed approach.

## 4.3 Qualitative Comparison

To further underscore the perceptual quality of our results, we present several illustrative cases comparing different image compression methods in Fig. 1, 5 and 6. In Fig. 5, it is clearly observed

that traditional compression methods such as BPG [5] and MSE-optimized compression methods like ELIC [18] produce overly smooth images with significant loss of detail information. Among the perception-oriented optimization methods, ILLM [42] introduces numerous continuous and repetitive artifacts in the decoded images, and the details in HiFiC [38] are not as clear as those in our method though its bitrate is much higher than ours. Fig. 6 shows the results on high-resolution images, whose subjective performance is generally consistent with that on the Kodak. Evidently, the reconstructed images using our method exhibit richer visual details, and less artifact while utilizing fewer or comparable bits.

## 4.4 Ablation Studies

We conduct extensive ablation studies for our proposed network architecture on the Kodak dataset. By replacing the syntax driven feature fusion module with direct element-wise addition, the model fuses information directly without syntax guided adaptive fusion(*w/o* SFF). By replacing the consistent boundary with random noise and inferring by pre-trained model parameters without online fine-tuning, the optimization during inference is moved (*w/o* Infer. Optim.). We do the above substitutions in turn and observe a performance drop as Table 2 shows, though the model sizes are kept almost the same. Hence, all of the components in our design contribute to performance improvement.

**Table 2: Average BD-rates of the ablation studies.**

| *w/* SFF | *w/* Infer. Optim. | Distortion | Perception | Average |
|---|---|---|---|---|
| ✓ | ✓ | — | — | — |
| ✓ | ✗ | -0.6501 | 1.7797 | 0.5648 |
| ✗ | ✗ | 0.8688 | 2.4148 | 1.6418 |

## 5 CONCLUSION

In this work, a novel consistency guided diffusion model with neural syntax is proposed, introducing a diffusion model for perceptual image compression. The consistency guidance architecture, neural syntax driven mechanism and inference time optimization strategy lead the diffusion model to stably reconstruct high-quality images, making the final output excellent in terms of perception and fidelity. Experimental evaluation shows the superiority of our proposed methods.

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
