# OpenReview forum: "Consistency Guided Diffusion Model with Neural Syntax for Perceptual Image Compression"
_acmmm.org/ACMMM/2024/Conference — MM2024 Poster_

### Official Review · Reviewer_AMhg · 2024-05-24

**Rating:** 2
**Confidence:** 4

**Summary:**

The authors propose an image compression pipeline which features a novel consistency guided diffusion model postprocessing step.
The consistency guided diffusion model 1) reduces distortion by using the initial autoencoder reconstruction as both unet-input and novel intermediate feature conditioning and 2) prevents reconstruction ambiguity and generative artifacts by injecting features from the original image via a novel syntax feature fusion strategy.
These proposed changes result in more faithful reconstructions with less artifacts compared to other autoencoder-based and generative compression codecs.

**Strengths:**

* The proposed syntax feature fusion module is interesting and novel. I especially like the idea to use features extracted from initial autoencoder as well as syntax information from the source image to guide the diffusion process.
* The authors perform many experiments and ablations to validate the performance of their method.

**Limitations:**

* Unconvincing results. While the proposed methods performs better in some quantitative metrics, performance in other metrics is comparable or worse. This would be ok with significantly better subjective perceptual quality reconstructions, yet the authors do not perform a user study to assess this, and the provided visual comparisons do not show significantly better improvements (at least in my opinion).
* Relatively weak related works section. Previous works HFD [1] and DIRAC [2] also have a similar diffusion post-processing strategy, and the differences to these works should be elaborated more in this section.
* Similarly, the method should also be compared with HFD [1] as reconstructions on the Kodak dataset are published by the authors (http://theis.io/hifidiff//).

[1] Hoogeboom, Emiel, et al. "High-fidelity image compression with score-based generative models." arXiv preprint arXiv:2305.18231 (2023).

[2] Ghouse, Noor Fathima Khanum Mohamed, et al. "Neural image compression with a diffusion-based decoder." (2022).

**Suitability:**

3

---

### Official Review · Reviewer_Zrvk · 2024-05-26

**Rating:** 4
**Confidence:** 4

**Summary:**

This manuscript proposes a strategy to reduce distortion introduced in lossy image compression trough neural approaches (e.g., autoencoders) based on duffision models. Standard diffusion models suffers of two issues that may augment the degradation, hence, the main contribution of the manuscript lies in Consistency Guided Diffusione Models (CGDM) and Syntax driven feature fusion (SFF) to mitigate such problems.

**Strengths:**

* The paper is well-written and easy to read, while the proposed approach is well-motivated.

* Experiments section presents a quantitative and qualitative comparison which suggest the proposed method is promising. Such experiments are conducted on two popular state-of-art dataset, i.e. Kodak and CLIC.

**Limitations:**

* The manuscript focuses on image compression, which means it presents a unimodal characteristics.

* The manuscript does not consider the recent INR-based method for image compression which should be cited and also taking in consideration for enforcing the comparison section. Here, the most recent papers:

    * Y. Strümpler, J. Postels, R. Yang, L. V. Gool, and F. Tombari, “Implicit neural representations for image compression,” European Conference on Computer Vision. Springer, 2022.

    * E. Dupont, H. Loya, M. Alizadeh, A. Golinski, Y. W. Teh, and A. Doucet, “COIN++: Neural compression across modalities,” Transactions on Machine Learning Research, 2022.

    * L. Catania and D. Allegra, “Nif: A fast implicit image compression with bottleneck layers and modulated sinusoidal activations,” ACM International Conference on Multimedia, 2023

    * T. Ladune, P. Philippe, F. Henry, G. Clare, and T. Leguay, “Cool-chic: Coordinate-based low complexity hierarchical image codec,” International Conference on Computer Vision, 2023



* The novelty is on the average, as the authors employ many SOTA methods for the proposed framework.

* The use of BD-Rate to prove the validity of the proposed approach should be better explained. Table 1 is not of immediate understanding.

* It is not clear if the source code will be made available after the publication. This is critical to give the opportunity to replicate the same experiments and allowing to properly compare with the proposed approach with new ones.

* Figure 5 and 6 are intended to show a qualitative comparison. However, both focus on a texturized area without strong edge and geometrical shape. I recommend to replace one of the Figure with something like the bottom row of the figure 4 of the supplementary material, in order to show the behavior in presence of very high frequencies.

------------
Overall, at the current state the manuscript presents some limitations but, in my opinion it is still acceptable and interesting for the ACMMM audience

**Suitability:**

2

---

### Official Review · Reviewer_kMQj · 2024-06-02

**Rating:** 4
**Confidence:** 3

**Summary:**

This paper proposes a diffusion based post-processing network to enhance the perceptual quality for the compressed images, which are coded with end-to-end image codec.
This paper proposes consistency guided diffusion model (CGDM) to improve the quality of the decoded image, preserving the style-like information with the original image.
Moreover, this paper also proposes a syntax driven feature fusion strategy to provide prior information of the original image, alleviating the ambiguity of the processed image.
Based on the experimental results, performance improvements could be observed on various test datasets and metrics.

**Strengths:**

This paper considers the ambiguity of the existing diffusion based image post-processing model, which only make use of the information from distorted images. The CGDM and syntax driven feature fusion strategy are proposed to tackle this problem. Performance improvements could be observed.

**Limitations:**

1. In table 1, there are various BD-rate numbers, evaluated under various metrics (various metrics for both distortion and perception). There are some very large numbers and the BD-rates of the distortion and perception are averaged directly. It could be more reasonable to provide the BD-rate results for every dataset under specific evaluation metric, for better understanding.
2. What's the computational complexity of the proposed model, compared with the learned codec baselines in this paper.
3. How the BD-rate is calculated for some models in table 1, since there are only 3 performance points for some methods (HIFIC and CDC), according to figure 4. Generally, at least, 4 points are required to calculate the BD-rate.
4. Regarding ablation study, it is interesting to compare the performance of the detailed techniques in "Inference Time Optimization", especially the performance of "Sample-Adaptive Continuous Online Optimization".

**Suitability:**

3

---

### Meta-Review · Area_Chair_2SiU · 2024-07-07

**Recommendation:** Accept (Poster)
**Confidence:** 4

**Metareview:**

The paper proposed a novel image compression pipeline composed of an end-to-end autoencoder plus a consistency-guided diffusion model, which works as a post-processing step.  Although the idea of using diffusion as post-processing for an autoencoder-like image compression framework is not completely new, the paper advances the state-of-the-art by considering the ambiguity of the existing diffusion-based and using a syntax-driven feature fusion strategy.  During the review phase, the reviewers raised concerns with regard to the novelty of the work (since there are other works on diffusion as post-processing for image compression), the computational complexity of the model, the lack of a further discussion of related work, more comparison (in particular with HFD), and lack of subjective study.  During the rebuttal, the authors answered most of the concerns, provided a small-scale subjective study, and compared their work with HFD regarding BD-Rate, showing better performance.

Overall, although the reviewers do not fully agree with the ratings, the paper can bring an interesting discussion to the conference.